# Neurologic manifestations in hospitalized patients with COVID-19 in Mexico City

**Fernando Daniel Flores-Silva**[1‡], **Miguel García-Grimshaw**[1‡], **Sergio Iván Valdés-Ferrer**[1,2‡], **Alma Poema Vigueras-Hernández**[1], **Rogelio Domínguez-Moreno**[1], **Dioselina Panamá Tristán-Samaniego**[1], **Anaclara Michel-Chávez**[1], **Alejandra González-Duarte**[1], **Felipe A. Vega-Boada**[1], **Isael Reyes-Melo**[1], **Amado Jiménez-Ruiz**[3], **Oswaldo Alan Chávez-Martínez**[1], **Daniel Rebolledo-García**[1], **Osvaldo Alexis Marché-Fernández**[1], **Samantha Sánchez-Torres**[1], **Guillermo García-Ramos**[1], **Carlos Cantú-Brito**[1]*, **Erwin Chiquete**[1]*

**1** Department of Neurology and Psychiatry, Instituto Nacional de Ciencias Médicas y Nutrición Salvador Zubirán, Mexico City, Mexico, **2** Center for Biomedical Science, Feinstein Institute for Medical Research, Manhasset, NY, United States of America, **3** Stroke, Dementia & Heart Disease Laboratory, Western University, London, ON, Canada

‡ These authors contributed equally to this work and merit first authorship on this work.
* erwin.chiquetea@incmnsz.mx (EC); carloscantu_brito@hotmail.com (CCB)

**Data Availability Statement:** All relevant data are within the paper and its Supporting Information file.

## Abstract

### Background

The coronavirus disease 2019 (COVID-19) is a systemic entity that frequently implies neurologic features at presentation and complications during the disease course. We aimed to describe the characteristics and predictors for developing in-hospital neurologic manifestations in a large cohort of hospitalized patients with COVID-19 in Mexico City.

### Methods

We analyzed records from consecutive adult patients hospitalized from March 15 to June 30, 2020, with moderate to severe COVID-19 confirmed by reverse transcription real-time polymerase chain reaction (rtRT-PCR) for the severe acute respiratory syndrome coronavirus 2 (SARS-CoV-2). Neurologic syndromes were actively searched by a standardized structured questionnaire and physical examination, confirmed by neuroimaging, neurophysiology of laboratory analyses, as applicable.

### Results

We studied 1,072 cases (65% men, mean age 53.2±13 years), 71 patients had pre-existing neurologic diseases (diabetic neuropathy: 17, epilepsy: 15, history of ischemic stroke: eight, migraine: six, multiple sclerosis: one, Parkinson disease: one), and 163 (15.2%) developed a new neurologic complication. Headache (41.7%), myalgia (38.5%), dysgeusia (8%), and anosmia (7%) were the most common neurologic symptoms at hospital presentation. Delirium (13.1%), objective limb weakness (5.1%), and delayed recovery of mental status after sedation withdrawal (2.5%), were the most common new neurologic syndromes. Age, headache at presentation, preexisting neurologic disease, invasive mechanical ventilation, and

**Funding:** The author(s) received no specific funding for this work.

**Competing interests:** The authors have declared that no competing interests exist.

neutrophil/lymphocyte ratio ≥9 were independent predictors of new in-hospital neurologic complications.

## Conclusions

Even after excluding initial clinical features and pre-existing comorbidities, new neurologic complications in hospitalized patients with COVID-19 are frequent and can be predicted from clinical information at hospital admission.

## Introduction

In December 2019, an outbreak of pneumonia of unknown etiology emerged in Wuhan City, Hubei Province of China [1]. By January 7, 2020, a new strain of β-coronavirus named severe acute respiratory syndrome coronavirus 2 (SARS-CoV-2) was identified as the etiological agent of coronavirus disease 2019 (COVID-19) [2]. In Mexico, the first official case was reported on February 28, 2020 [3], and on March 11, 2020, the World Health Organization (WHO) declared COVID-19 a global pandemic [4].

From the initial clinical descriptions of COVID-19, there has been accruing evidence on the potential neurologic involvement of SARS-CoV-2, manifested by symptoms such as anosmia, dysgeusia, muscle pain, and headache during the early course of the disease [5–8]. Although these are non-specific symptoms, the neurologic symptoms have led to many hypotheses on how the virus reaches the nervous system, including its potential entry via the olfactory groove or the bloodstream [9–12]. There are also well-documented cases and case series describing a direct neurologic viral involvement and some post-infectious manifestations [11].

Currently, the Latin American region accounts for most of the currently active cases of COVID-19 [13]. Worldwide, there has been an increasing awareness of the long-term neurologic consequences for the surviving patients [14, 15]. The description of neurologic manifestations in COVID-19 patients is limited to a few series, and to the best of our knowledge [16–19], large series describing this disease's neurologic implications in Latin America are lacking. We aimed to describe the characteristics and clinical predictors for developing in-hospital neurologic manifestations in a large cohort of hospitalized COVID-19 patients of Mexico City. This is the first cross-sectional report from a large study aimed to characterize neurologic complications of COVID-19 in the long term.

## Methods

### Study design, patient selection, and setting

This prospective, cross-sectional, observational study was conducted at the *Instituto Nacional de Ciencias Médicas y Nutrición Salvador Zubirán*, a COVID-19-designated tertiary-care referral hospital in Mexico City. Since the very beginning, standardized diagnostic and care protocols were implemented. Signed informed consent was obtained from patients or next-of-kin on admission. Clinical information was captured in a database derived from electronic medical records designed for patients' care and research. The first patient included in the research database was recorded on March 15, 2020. The Neuro-COVID-19 research team accessed this prospective database first in May 01, 2020. The *Instituto Nacional de Ciencias Médicas y Nutrición Salvador Zubirán* Research and Ethics Committees approved the study (reference number

CAI-3497-20-20-1). For the present report, we analyzed consecutive hospitalized adult (≥18 years) patients hospitalized from March 15 to June 30, 2020, who had confirmed COVID-19 by positive SARS-CoV-2 reverse transcription real-time polymerase chain reaction (rtRT-PCR) in respiratory fluids from nasal swabs and chest CT scan. For the purpose of the present analysis, we excluded patients with negative rtRT-PCR for SARS-CoV-2, non-hospitalized ambulatory patients, and those discharged or transferred to other hospitals within the first 24 hours after admission. The present report aimed to describe differences in demographics, clinical presentation, and disease severity in patients who developed new neurologic events diagnosed during their hospital stay with those patients who did not. For these purposes, we defined patients with new neurologic events as those who suffered delirium, delayed recovery from medical sedation, myopathies, peripheral nerve syndromes, seizures, encephalopathy, stroke (either hemorrhagic or ischemic), and any other established new neurologic diagnosis not previously identified at hospital admission.

## Data collection

We reviewed electronic medical records, laboratory findings, and radiologic examinations from all patients. Data collection included baseline demographic characteristics and comorbidities, including the history of a diagnosed cardiovascular, pulmonary, neurologic, or autoimmune disease. We also recorded symptoms at presentation (i.e., fever, rhinorrhea, cough, headache, anosmia, dysgeusia, myalgia, arthralgia, nausea, vomiting, and diarrhea); time from symptom onset to hospital presentation; vital signs on admission; laboratory findings including complete blood count, blood chemistry (renal and liver function tests, creatine kinase [CK] and lactate dehydrogenase), inflammatory response biomarkers (serum ferritin, D-dimer, C-reactive protein, and fibrinogen); arterial blood gas analysis; chest CT scan findings; days from admission to development of neurologic events; requirement of intensive care unit (ICU) or requirement of invasive mechanical ventilation (IMV); days of in-hospital stay; and hospital outcome. The information was recorded using a standardized case report format. Two physicians reviewed all data, and a third researcher adjudicated any difference in interpretation between the two primary reviewers.

## Definitions

We defined an altered mental status using a Glasgow coma score ≤13 points at presentation. Delirium was defined as fluctuation in mental status. Delayed recovery of mental status after sedation was defined as persistent altered mental status lasting more than four half-lives after discontinuation of sedative drugs. Acute ischemic or hemorrhagic stroke was diagnosed by brain CT, MRI, or both. Encephalitis was defined as persistent altered mental status in patients with an abnormal MRI and inflammatory changes in the CSF. Weakness was defined clinically. HyperCKemia was defined as a CK level ≥ 200 U/L, according to our laboratory upper limit of normality. Myopathy was defined as the presence of myalgia at admission and a CK level ≥ 200 U/I. Obesity was defined as a body mass index (BMI) ≥ 30. In all patients, a chest CT was performed and evaluated by experienced radiologists. A visual semi-quantitative classification of the severity of lung involvement assessing total pulmonary consolidation/ground-glass opacities as mild (extension of the disease ≤20% of the pulmonary parenchyma), moderate (20–50%), or severe (> 50%) was used. The $PaO_2/FiO_2$ ratio was calculated using the estimated fractions of inspired oxygen provided by each oxygen delivery device, according to the oxygen flow rate (L/min), when the first arterial blood gas measurement was obtained. Acute respiratory distress syndrome (ARDS) was categorized according to the Berlin definition into mild, moderate, and severe [20]. The neutrophil/lymphocyte ratio was categorized as normal/

mild (1–9), moderate (9–18), and severe (>18) [21]. On admission, we calculated illness severity according to the National Early Warning Score (NEWS) 2 and the disease progression risk with the comorbidities, age, lymphocytes and lactic dehydrogenase (LDH) (CALL) scoring model [22, 23].

## Statistical analyses

Categorical variables are reported as frequencies and proportions, and continuous variables are described as median with interquartile range (IQR) or as mean with standard deviation (SD). Analyses of differences for independent groups between categorical variables were performed with the $X^2$ or Fisher's exact tests, and for continuous variables with the Student's $t$-test or Mann-Whitney $U$ test, as appropriate. We performed a binary logistic regression analysis to determine predictors for the development of in-hospital neurologic manifestations, including all the independent co-variables based on biological plausibility and those with $p \leq 0.1$. The model adjustment was evaluated by the Hosmer-Lemeshow test, which was considered reliable when the p value was $\geq 0.20$. Odds ratios (OR) with 95% CI were calculated. All values were two-tailed and considered significant when the $p < 0.05$. All statistical analyses were performed with IBM SPSS Statistics, version 26 (IBM Corp., Armonk, NY, USA).

## Results

During the study period, a total of 1,235 patients with COVID-19 were hospitalized in our center. We excluded 163 cases for the following reasons: 107 had negative (one or multiple) SARS-CoV-2 rtRT-PCR tests results, and 56 were discharged or transferred to other hospitals within the first 24 hours after admission. We studied 1072 cases, 375 (35%) women, and 697 (65%) men, with a mean age of 53.2±13 years (Table 1). There were no demographic differences between patients who developed in-hospital neurologic events and those who did not. The most common comorbidity was obesity (46.5%), followed by hypertension (30.3%) and diabetes (27.9%).

Seventy-one patients had pre-existing neurologic disease: diabetic neuropathy in 17, epilepsy in 15, eight with a history of ischemic stroke (none hemorrhagic), six with migraine, and one with multiple sclerosis, and Parkinson's disease, respectively. Pre-existing neurologic disorders were more frequent in those who developed neurologic events during hospitalization (12.3% vs. 5.6%; $p = 0.002$). The median number of days from symptoms onset to admission was seven without differences between groups; the most frequent non-neurologic symptoms were dyspnea in 923 (86.1%), fever in 896 (83.6%), and cough in 834 (77.8%); overall, there were no differences of presenting symptoms within groups.

Hemoglobin and lymphocyte levels were significantly lower in patients who developed in-hospital neurologic manifestations. In contrast, the total neutrophil count was higher in this group. Although there were statistical differences in creatinine and blood urea nitrogen levels, both fell within normal limits. Plasma liver enzymes activity was within normal limits in both groups, but patients with neurologic manifestations had significantly lower albumin levels. C-reactive protein, D-dimer, and fibrinogen levels were significantly higher among patients with in-hospital events. Findings of arterial blood gas analysis were comparable between groups; however, patients with neurologic manifestations had higher oxygen requirements (46.8± 6.2 vs. 39.3±16.7 $p < 0.001$), breathing rate (32±10 vs. 29±8 $p < 0.001$), and heart rate 105±18 vs. 102±17 $p = 0.021$).

On admission, patients who developed in-hospital neurologic manifestations had increased disease severity as evidenced by lower $PaO_2/FiO_2$ ratios (164.9±81.4 vs. 207.9±98.6; $p < 0.001$), higher neutrophil/lymphocyte ratio (12.87, IQR: 8.44–20 vs. 8.5 IQR: 5.2–14.92; $p < 0.001$),

**Table 1. Characteristics of the patients hospitalized with moderate to severe COVID-19.**

| | All patients (n = 1,072) | Without neurologic events (n = 909) | With neurologic events (n = 163) | p value |
|---|---|---|---|---|
| Demographics | | | | |
| Age, mean (±SD), years | 53.2 (13.7) | 53.1 (13.5) | 54.1 (14.9) | 0.389 |
| Male, n (%) | 697 (65) | 582 (64) | 115 (70.6) | 0.108 |
| Comorbidities | | | | |
| Diabetes, n (%) | 299 (27.9) | 254 (27.9) | 45 (27.6) | 0.93 |
| Hypertension, n (%) | 325 (30.3) | 270 (29.7) | 55 (33.7) | 0.302 |
| Smoking, n (%) | 157 (14.6) | 137 (15.1) | 20 (12.3) | 0.352 |
| Obesity*, n (%) | 499 (46.5) | 428 (47.1) | 71 (43.6) | 0.406 |
| BMI, mean (±SD) | 30.1 (5.8) | 30.2 (5.8) | 29.6 (5.9) | 0.219 |
| HIV infection, n (%) | 12 (1.1) | 8 (0.9) | 4 (2.5) | 0.095 |
| Pulmonary disease, n (%) | 42 (3.9) | 33 (3.6) | 9 (5.5) | 0.252 |
| Cardiovascular disease, n (%) | 57 (5.3) | 49 (5.4) | 8 (4.9) | 0.798 |
| Neurologic disease, n (%) | 71 (6.6) | 51 (5.6) | 20 (12.3) | 0.002 |
| Autoimmune disease, n (%) | 69 (6.4) | 56 (6.2) | 13 (8) | 0.385 |
| Non-neurologic symptoms | | | | |
| Length from symptoms onset to hospital admission, median (IQR), days | 7 (6–10) | 8 (6–10.5) | 7 (6–10) | 0.418 |
| Dyspnea, n (%) | 923 (86.1) | 770 (84.7) | 153 (93.9) | 0.004 |
| Fever, n (%) | 896 (83.6) | 763 (84.3) | 133 (79.8) | 0.457 |
| Cough, n (%) | 834 (77.8) | 708 (77.9) | 126 (77.3) | 0.868 |
| Diarrhea, n (%) | 188 (17.5) | 154 (16.9) | 34 (20.9) | 0.226 |
| Rhinorrhea, n (%) | 157 (14.6) | 129 (14.2) | 28 (17.2) | 0.321 |
| Chest pain, n (%) | 142 (13.2) | 124 (13.6) | 18 (11) | 0.368 |
| Vomiting, n (%) | 59 (5.5) | 48 (5.3) | 11 (6.7) | 0.449 |
| Joints pain, n (%) | 345 (32.2) | 283 (31.1) | 62(38) | 0.082 |
| Laboratory findings | | | | |
| Hemoglobin, mean (±SD), g/dL | 15.2 (1.9) | 15.2 (1.9) | 14.6 (2.1) | 0.01 |
| WBC, median (IQR), $10^9$/L | 8.3 (5.9–11.6) | 8.1 (5.9–11.1) | 9.8 (6.7–13.7) | 0.001 |
| Neutrophils, median (IQR), $10^9$/L | 6.8 (4.6–10.1) | 6.6 (4.5–9.6) | 8.7 (5.5–12.2) | < 0.001 |
| Lymphocytes, median (IQR), $10^9$/L | 0.75 (0.52–1.02) | 0.78 (0.54–1.04) | 0.65 (0.45–0.87) | < 0.001 |
| Platelets, median (IQR), $10^9$/L | 226 (181–300) | 226 (181–296) | 228 (181–328) | 0.277 |
| Glucose, median (IQR), mg/dL | 122 (105–173) | 121 (105–173.5) | 131 (107–175) | 0.202 |
| Creatinine, median (IQR), mg/dL | 0.94 (0.77–1.2) | 0.93 (0.76–1.17) | 1.01 (0.81–1.31) | 0.012 |
| Blood urea nitrogen, median (IQR), mg/dL | 16.2 (11.8–24.2) | 15.7 (11.5–23) | 20.4 (13.6–29.2) | < 0.001 |
| Sodium, mean (±SD), mmol/L | 135 (4.5) | 134.9 (4.4) | 135.5 (4.9) | 0.097 |
| Potassium, mean (±SD), mmol/L | 4.1 (0.67) | 4.05 (0.68) | 4.08 (0.57) | 0.525 |
| ALT, median (IQR), U/L | 36.4 (24.1–55.5) | 36.3 (24–55.6) | 37.8 (24.4–55.1) | 0.901 |
| AST, median (IQR), U/L | 42.5 (30–62.8) | 42 (29.6–62.8) | 46.6 (31.4–61.8) | 0.459 |
| Albumin, mean (±SD), g/dL | 3.6 (0.5) | 3.6 (0.5) | 3.4 (0.5) | < 0.001 |
| LDH, median (IQR), U/L | 385 (298–504) | 377 (295–498) | 422 (342–554) | 0.001 |
| CK, median (IQR), U/L | 110 (57–221) | 98 (56–216) | 125 (63–239) | 0.065 |
| HyperCKemia*, n (%) | 281 (26.2) | 230 (26.6) | 51 (31.9) | 0.169 |
| C-reactive protein, mean (±SD), mg/dL | 16.4 (10) | 14.9 (7.2–23) | 19.2 (12.5–27.2) | <0.001 |
| Ferritin, median (IQR), ng/dL | 629 (327–1,086) | 614 (324–1,079) | 684 (372–1,189) | 0.142 |

*(Continued)*

**Table 1.** (Continued)

| | All patients (n = 1,072) | Without neurologic events (n = 909) | With neurologic events (n = 163) | p value |
|---|---|---|---|---|
| Fibrinogen, mean (±SD), mg/dL | 691.9 (212) | 685.3 (213.2) | 727.3 (202.5) | 0.026 |
| D-dimer, median (IQR), ng/dL | 829 (524–1323) | 811 (508–1,289) | 944 (662–1,782) | 0.004 |
| Serum lactate, median (IQR), U/L | 1.1 (1.1–2.0) | 1.4 (1.1–2.0) | 1.5 (1.1–2.1) | 0.145 |

ALT, alanine aminotransferase; AST, aspartate aminotransferase; BMI, body mass index; CK, creatine kinase; HIV, human immunodeficiency virus; IQR, interquartile range; LDH, Lactate dehydrogenase; SD, standard deviation; WBC, white blood cells.

*Defined as a body mass index ≥30.

** Defined as a creatine kinase ≥200 U/L.

and NEWS2 score (7.55±1.77 vs. 6.87±1.87; $p < 0.001$), compared with those patients who did not develop neurologic complications. Lung damage on admission (measured by chest CT) was also more significant among those patients developing in-hospital neurologic dysfunction (Table 2).

Neurologic manifestations and in-hospital outcomes can be seen in Table 3. Headache was the most common neurologic symptom on admission (41.7%), followed by myalgia (38.5%), dysgeusia (8%), and anosmia (7%); however, there were no differences in those symptoms between groups. Eighteen patients presented with altered mental status, being more frequent in patients who developed in-hospital neurologic events (6.7% vs. 0.8% $p < 0.001$). The most common neurologic event during hospitalization was delirium, observed in 140 (13.1%) patients, weakness in 55 (5.1%), and delayed recovery of mental status after sedation withdrawal in 27 (2.5%).

Nine patients developed seizures, eight of which were related to metabolic abnormalities. One patient developed myoclonic status epilepticus after an episode of hypoxia-related cardiac arrest with bilateral watershed ischemic lesions in the occipital, parietal and frontal lobes and died 48 hours after the event. Two patients developed encephalitis; both tested negative for SARS-CoV-2 rtRT-PCR in CSF. Nine patients developed stroke events, one of them was diagnosed at presentation; six ischemic, all within the anterior circulation (none within reperfusion therapy timeframe), and three with intracerebral hemorrhage. Four patients developed metabolic-related myoclonic jerks.

Regarding the timing of the events, seizures (median 15 days, IQR 7–27) developed earlier during the hospitalization, followed by delirium (median 16 days, IQR 7.5–21), weakness (median 20 days, IQR 15–25), and stroke (median 24 days, IQR 6–30), delayed recovery of mental status after sedation withdrawal (median 17, IQR 9–27), and encephalitis (median 9.5, IQR 2–17) (Fig 1). Neuroimaging studies were performed in only 19% of patients with neurologic symptoms; CSF analysis and EEG where routinely performed in only 5.5% and 3.7% of cases, respectively. Testing for SARS-COV-2 rtRT-PCR in CSF was negative in all patients. Admission to the ICU and the use of IMV were more frequent in patients with neurologic events, and the length of in-hospital stay was longer in that group. Although patients with neurologic events had increased disease severity and higher IMV requirements, mortality rates were similar between groups (25.5% vs. 19.6%; $p = 0.115$).

In a multivariate analysis adjusted for relevant covariables (i.e, chest CT findings, prognosis scores, symptoms at admission, biomarkers, length from symptoms onset to hospital admission, and length of stay): age, headache on admission, preexisting neurologic disease, the need for IMV, and a neutrophil/lymphocyte ratio ≥9 were independent predictive factors for the

**Table 2. Disease severity at hospital admission in patients hospitalized with moderate to severe COVID-19.**

|  | All patients (n = 1,072) | Without neurologic events (n = 909) | With neurologic events (n = 163) | *p* value |
|---|---|---|---|---|
| Chest CT severity |  |  |  | < 0.001 |
| Mild, n (%) | 92 (8.6) | 87 (9.6) | 5 (3.1) |  |
| Moderate, n (%) | 384 (35.9) | 346 (38.1) | 38 (23.5) |  |
| Severe, n (%) | 594 (55.5) | 475 (52.3) | 119 (73.5) |  |
| ARDS severity |  |  |  | < 0.001 |
| Without, n (%) | 151 (14.3) | 139 (15.5) | 12 (7.4) |  |
| Mild, n (%) | 333 (31.5) | 299 (33.4) | 34 (21) |  |
| Moderate, n (%) | 411 (38.8) | 337 (37.6) | 74 (45.7) |  |
| Severe, n (%) | 163 (15.4) | 121 (13.5) | 42 (25.9) |  |
| Neutrophil/lymphocyte ratio |  |  |  | < 0.001 |
| Normal-Mild, n (%) | 522 (48.7) | 478 (52.6) | 44 (27) |  |
| Moderate, n (%) | 328 (30.6) | 261 (28.7) | 67 (41.1) |  |
| Severe, n (%) | 222 (20.7) | 170 (18.7) | 52 (31.9) |  |
| CALL score |  |  |  | 0.016 |
| Low risk, n (%) | 379 (35.4) | 311 (34.2) | 68 (41.7) |  |
| Intermediate risk, n (%) | 424 (39.6) | 376 (41.4) | 48 (29.4) |  |
| High risk, n (%) | 269 (25.1) | 222 (24.4) | 47 (28.8) |  |
| NEWS2 score |  |  |  | 0.003 |
| Low risk, n (%) | 120 (11.2) | 114 (12.5) | 6 (3.7) |  |
| Medium risk, n (%) | 252 (23.5) | 215 (23.9) | 37 (22.7) |  |
| High risk, n (%) | 700 (65.3) | 580 (63.8) | 120 (73.6) |  |

ARDS, acute respiratory distress syndrome; $FiO_2$, fraction of inspired oxygen; IQR, interquartile range; NEWS2, National Early Warning Score 2; $PaO_2$, partial pressure of oxygen; SD, standard deviation.

development of new in-hospital neurologic events that were not present at hospital presentation (Table 4).

## Discussion

This is the largest single-center study describing the in-hospital neurologic manifestations of patients with confirmed COVID-19 in Latin America, a region where the pandemic has been particularly severe [13]. In this cohort, the incidence of hospital neurologic outcomes was 15.2%, a frequency similar to the 13.5% reported in a series of 4,491 hospitalized patients from four New York City hospitals [24] (Table 5). When including non-specific neurologic manifestations such as headache, anosmia, dysgeusia, and myalgia, this frequency increased to 69.3%, which is slightly higher than the reported in similar series with an overall frequency of 56.4% [16–19].

Similar to other series, most of our patients were males, and age was similar to that of the first neurologic series in China reported by Mao et al. [17] but younger than in others [16, 18, 19]. In all reported series, risk factors such as diabetes, hypertension, and obesity were high for a relatively young population, and we found these disorders to be slightly less frequent than the New York and Italian populations [18, 24], but similar to the rates reported by Romero-Sánchez et al. in Spain [16]. These contrasting results may be related to the different included age groups among the series. Comparable to the findings reported by Frontera et al. [24], we found that pre-existing neurologic diseases were related to the development of in-hospital neurologic manifestations. Other symptoms that were more frequent among patients with neurologic events in our study included muscle pain and altered mental status upon admission.

**Table 3. Neurologic manifestations and outcomes of the patients hospitalized with moderate to severe COVID-19.**

| | All patients (n = 1,072) | Without neurologic events (n = 909) | With neurologic events (n = 163) | p value |
|---|---|---|---|---|
| Neurologic manifestations at admission, n (%) | 697 (65) | 580 (63.8) | 117 (71.8) | 0.049 |
| Headache, n (%) | 447 (41.7) | 371 (40.8) | 76 (46.6) | 0.166 |
| Muscle pain, n (%) | 413 (38.5) | 338 (37.2) | 75 (46) | 0.033 |
| Myopathy, n (%) | 106 (9.9) | 85 (9.4) | 21 (12.9) | 0.164 |
| Anosmia, n (%) | 75 (7) | 64 (7) | 11 (6.7) | 0.893 |
| Dysgeusia, n (%) | 86 (8) | 72 (7.9) | 14 (8.6) | 0.772 |
| Dizziness, n (%) | 13 (1.2) | 9 (1) | 4 (2.5) | 0.121 |
| Altered mental status, n (%) | 18 (1.7) | 7 (0.8) | 11 (6.7) | < 0.001 |
| Glasgow coma scale, mean (±SD) | 14.92 (0.61) | 14.94 (0.56) | 14.75 (0.85) | 0.007 |
| In-hospital neurologic manifestations | 163 (15.2) | - | - | - |
| Delirium, n (%) | 140 (13.1) | - | 140 (85.9) | - |
| Delayed recovery of mental status after sedation, n (%) | 27 (2.5) | - | 27 (16.6) | - |
| Seizures, n (%) | 9 (0.8) | - | 9 (5.5) | - |
| Stroke, n (%) | 9 (0.8) | - | 9 (5.5) | - |
| Ischemic, n (%) | 6 (0.6) | - | 6 (3.7) | - |
| Hemorrhagic, n (%) | 3 (0.3) | - | 3 (1.8) | - |
| Encephalitis, n (%) | 2 (0.2) | - | 2 (1.2) | - |
| Weakness, n (%) | 55 (5.1) | - | 55 (33.7) | - |
| Outcomes | | | | |
| ICU admission/IMV, n (%) | 250 (23.3) | 131 (14.4) | 119 (73) | <0.001 |
| Interval from hospital admission to IMV (days), median (IQR) | 2 (1–4) | 2 (1–3) | 2 (1–4) | 0.005 |
| Interval from sedation withdrawal to IMV weaning (days), median (IQR) | 3 (1–5) | 1 (0–3) | 3 (1–6) | <0.001 |
| Days of in-hospital stay, median (IQR), days | 7 (4–12) | 6 (4–9) | 23 (14–32) | <0.001 |
| Death, n (%) | 264 (24.6) | 232 (25.5) | 32 (19.6) | 0.115 |

ICU, intensive care unit; IMV, invasive mechanical ventilation; IQR, interquartile range.

We found that factors associated with the development of new in-hospital neurologic events seem to be mostly related to disease severity, both in respiratory parameters (PaO$_2$/FiO$_2$ ratio, ARDS severity, and chest CT findings) and inflammatory markers (C-reactive protein, D-

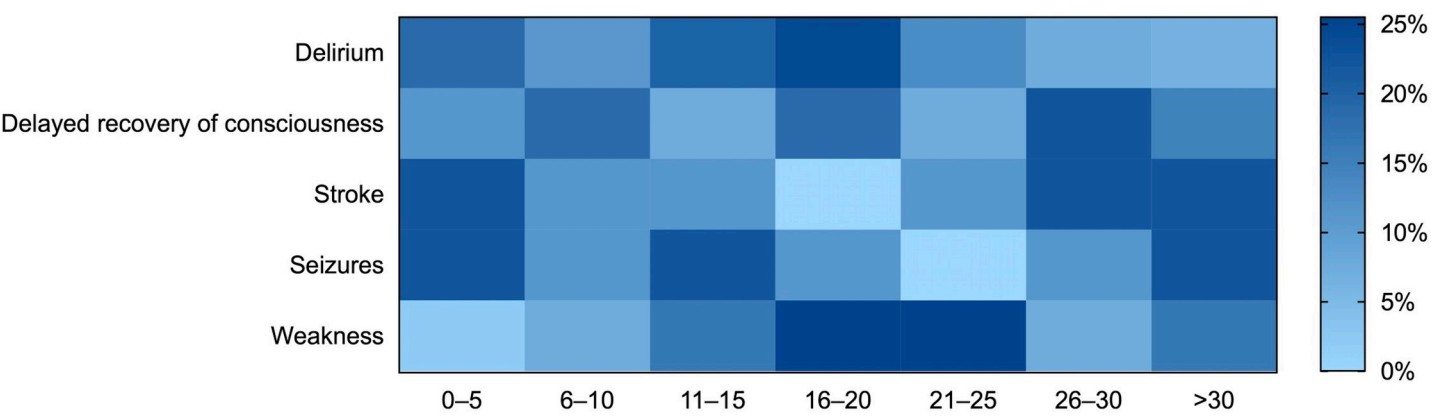

**Fig 1. Days from admission to the appearance of neurologic events.** Delayed recovery of consciousness was defined as a persistent altered mental status lasting longer than four half-lives after the withdrawal of sedative drugs.

**Table 4. Predictors for the development of new neurologic events during hospitalization in patients hospitalized with moderate to severe COVID-19***.

| Variable | Odds ratio (95% CI) | *p* value |
|---|---|---|
| Age, per year increment | 1.024 (1.007–1.040) | 0.005 |
| Preexisting neurologic disease | 5.188 (2.615–10.294) | < 0.001 |
| Headache at admission | 1.532 (1.020–2.301) | 0.04 |
| ICU admission | 20.591 (13.030–32.540) | < 0.001 |
| Neutrophil/lymphocyte ratio ≥ 9 points | 1.952 (1.267–3.010) | 0.002 |

ICU, intensive care unit.

* Model adjusted for sex, chest CT findings, comorbidities, non-neurologic symptoms at admission, anosmia, dysgeusia, D-dimer, and $PaO_2/FiO_2$ ratio, length from symptoms onset to hospital admission, and length of stay.

dimer, and neutrophil index/lymphocyte). Interestingly, headache on admission remained a factor associated with this outcome in the multivariate analysis; however, the frequency of headache in our study was higher than in other studies [16–19, 25, 26].

In contrast to the findings reported in New York City [24], despite the differences in disease severity in our population, the development of in-hospital neurologic events was not associated with a high mortality. Notably, these rates were lower than those reported by them (19.6% vs. 35%), but the length of hospitalization was two-fold. This may be directly related to disease severity, where 73% of our patients required IVM compared to 40% of theirs.

It is interesting that despite the younger age of patients, the reported frequency of major neurologic events such as altered mental status, cerebrovascular disease, and seizures remain relatively constant throughout most series. Regarding the timing for developing a neurologic event, we observed they had a wide array of distribution. Weakness and altered mental status

**Table 5. Main neurologic manifestations reported among series of hospitalized patients with COVID-19.**

| Neurologic syndromes | Mexico (n = 1,072) | Spain [16] (n = 841) | Turkey [19] (n = 239) | China [17] (n = 214) | Italy [18] (n = 213) | Total, n (%) |
|---|---|---|---|---|---|---|
| Any neurologic symptom, n (%) | 743 (69.3) | 483 (57.4) | 83 (34.7) | 78 (36.4) | 64 (30) | 1445/2559 (56.4) |
| Central nervous system | | | | | | |
| Headache, n (%) | 447 (41.7) | 119 (14.1) | 64 (26.7) | 28 (13.1) | 10 (4.6) | 668/2559 (26.1) |
| Dizziness, n (%) | 13 (1.2) | 51 (6.1) | 16 (6.7) | 36 (16.8) | 3 (1.4) | 119/2559 (4.6) |
| Altered mental state, n (%) | 167 (15.6) | 165 (19.6) | 23 (9.6) | 16 (7.5) | 11 (5.1) | 386/2559 (15.1) |
| Seizures, n (%) | 9 (0.8) | 6 (0.7) | NR | 1 (0.5) | 6 (2.8) | 22/2320 (0.9) |
| Cerebrovascular disease*, n (%) | 9 (0.8) | 14 (1.7) | 9 (3.8) | 6 (2.8) | 4 (1.9) | 33/2559 (1.29) |
| Encephalitis, n (%) | 2 (0.2) | 1 (0.1) | NR | NR | 1 (0.5) | 4/2106 (0.2) |
| Movement disorders, n (%) | 4 (0.3) | 6 (0.7) | NR | NR | NR | 10/1893 (0.5) |
| Peripheral nervous system | | | | | | |
| Anosmia, n (%) | 75 (7) | 41 (4.9) | 18 (7.5) | 11 (5.1) | 13 (6.1) | 158/2559 (6.2) |
| Dysgeusia, n (%) | 86 (8) | 52 (6.2) | 16 (6.7) | 12 (5.6) | 6 (2.8) | 172/2559 (6.7) |
| Muscle pain, n (%) | 413 (38.5) | 145 (17.2) | 36 (15.1) | NR | 30 (9.3) | 624/2345 (26.6) |
| Myopathy, n (%)** | 106 (9.9) | 26 (3.1)*** | NR | 23 (10.7) | 10 (4.7) | 165/2320 (7.1) |
| Weakness, n (%) | 55 (5.1) | 26 (3.1) | NR | NR | NR | 70/1893 (3.7) |

CK, creatine kinase; NR, not reported.

*Ischemic or hemorrhagic stroke.

**Defined as the presence of myalgia at admission and a creatine kinase level ≥ 200 U/L.

*** Creatine kinase value not defined.

had a delayed onset, probably related to disease severity, the length of hospitalization, and the requirement of IMV.

The mechanisms for the development of these neurologic manifestations remain incompletely understood. Since the discovery of SARS-CoV-2, many hypotheses have arisen, including its neuroinvasive potential via the olfactory groove or direct invasion of the nervous system via the bloodstream [9, 10, 12]. Alternatively, these findings may be secondary immunological mechanisms and the severe inflammatory state in response to the infection and severe hypoxemia promoted by critical illness and the comorbid conditions [7, 9–11, 27–31].

There are some limitations of the present study that should be considered, such as the low frequency of confirmatory neurologic studies given the limitations for patients' mobilization, making a precise neurologic phenotype in all patients partly elusive. Given the global severity of the pandemic and the requirement for available beds for their care, it is possible that the full spectrum of neurologic manifestations during hospitalization, as well as the potential future sequelae in the surviving patients, may not have been identified. We are currently performing a follow-up study on this cohort to complete the clinical evaluations and to characterize the neurologic syndromes in the long term. Some of the present study's strengths include that this is the first published cohort of Latin America, one of the regions with the highest active cases burden. We only included patients with confirmed SARS-CoV-2 pneumonia from various social classes, and the management was given in a hospital dedicated solely to COVID-19 care.

## Conclusion

In this study we found a high frequency of neurologic manifestations during hospitalization in COVID-19 patients suggesting a potentially high burden of short and long-term sequelae these for these patients. A precise monitoring strategy for neurologic outcomes in the convalescent stage is needed for all involved population sub-groups. This study represents the cross-sectional analysis of an ongoing longitudinal assessment of neurologic syndromes associated with COVID-19 in the Mexico City.

## Supporting information

**S1 File.**
(SAV)

## Acknowledgments

The authors are very thankful with the labor that all the health care workers of the *Instituto Nacional de Ciencias Médicas y Nutrición Salvador Zubirán* have done during this pandemic.

## Author Contributions

**Conceptualization:** Fernando Daniel Flores-Silva, Miguel García-Grimshaw, Felipe A. Vega-Boada, Amado Jiménez-Ruiz, Carlos Cantú-Brito, Erwin Chiquete.

**Data curation:** Fernando Daniel Flores-Silva, Miguel García-Grimshaw, Sergio Iván Valdés-Ferrer, Alma Poema Vigueras-Hernández, Dioselina Panamá Tristán-Samaniego, Anaclara Michel-Chávez, Alejandra González-Duarte, Felipe A. Vega-Boada, Isael Reyes-Melo, Amado Jiménez-Ruiz, Daniel Rebolledo-García, Osvaldo Alexis Marché-Fernández, Erwin Chiquete.

**Formal analysis:** Fernando Daniel Flores-Silva, Miguel García-Grimshaw, Rogelio Domínguez-Moreno, Carlos Cantú-Brito, Erwin Chiquete.

**Investigation:** Fernando Daniel Flores-Silva, Miguel García-Grimshaw, Sergio Iván Valdés-Ferrer, Alma Poema Vigueras-Hernández, Rogelio Domínguez-Moreno, Dioselina Panamá Tristán-Samaniego, Anaclara Michel-Chávez, Alejandra González-Duarte, Isael Reyes-Melo, Daniel Rebolledo-García, Erwin Chiquete.

**Methodology:** Fernando Daniel Flores-Silva, Miguel García-Grimshaw, Sergio Iván Valdés-Ferrer, Erwin Chiquete.

**Writing – original draft:** Fernando Daniel Flores-Silva, Miguel García-Grimshaw, Sergio Iván Valdés-Ferrer, Alma Poema Vigueras-Hernández, Oswaldo Alan Chávez-Martínez, Osvaldo Alexis Marché-Fernández, Samantha Sánchez-Torres, Guillermo García-Ramos, Carlos Cantú-Brito, Erwin Chiquete.

**Writing – review & editing:** Carlos Cantú-Brito, Erwin Chiquete.

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
