## [Decision Letter · Decision Letter 0]

14 Jan 2021

PONE-D-20-39539

Neurological manifestations in hospitalized patients with COVID-19 in Mexico City

PLOS ONE

Dear Dr. Chiquete,

Thank you for submitting your manuscript to PLOS ONE. After careful consideration, we feel that it has merit but does not fully meet PLOS ONE’s publication criteria as it currently stands. Therefore, we invite you to submit a revised version of the manuscript that addresses the points raised during the review process.

We look forward to receiving your revised manuscript.

Kind regards,

Tai-Heng Chen, M.D.

Academic Editor

PLOS ONE

2. Thank you for your ethics statement:

"Institutional Research and Ethics Committees approved the study (reference number

CAI-3497-20-20-1)."

3. Thank you for providing the date(s) when patient medical information was initially recorded. Please also include the date(s) on which your research team accessed the databases/records to obtain the retrospective data used in your study.

5. Please amend the manuscript submission data (via Edit Submission) to include author Daniel Rebolledo-García.

Reviewers' comments:

Reviewer's Responses to Questions

**Comments to the Author**

1. Is the manuscript technically sound, and do the data support the conclusions?

Reviewer #1: Yes

Reviewer #2: Yes

2. Has the statistical analysis been performed appropriately and rigorously? 

Reviewer #1: Yes

Reviewer #2: Yes

3. Have the authors made all data underlying the findings in their manuscript fully available?

Reviewer #1: Yes

Reviewer #2: Yes

4. Is the manuscript presented in an intelligible fashion and written in standard English?

Reviewer #1: Yes

Reviewer #2: Yes

5. Review Comments to the Author

Reviewer #1: Overall, a very nice & timely cross-sectional study that provides important insight in to the neurologic manifestations of the COVID-19. The manuscript is well written and precise. It certainly deserves publication.

I see no need for major revisions.

I have 2 suggestions for minor revisions: (1) I would recommend a more explicit statement of the study design be included in the methods section. That the study is a cross-sectional analysis is mentioned in the introduction but should be explicitly stated in the methods. (2) Although the comparison groups are evident from the Tables, it is not clear from the text of the manuscript. The addition of text in the methodology ("The study aims to describe differences in demographics, clinical presentation, and disease severity in those patients who experienced neurologic events and those who did not. For these purposes, we define patients with neurologic events as those who suffered delirium, delayed recovery...") or something similar. This addition will lend more clarity to the text.

Also, note that on p.7 of the submission, "Data Collection" section, there is a missing comma between pulmonary and neurological. Neurologic (rather than neurological - meaning pertaining to the study of or discipline of neurology) would be the preferred usage.

Reviewer #2: The authors have done a nice job in describing the neurological manifestations of COVID19 in Mexico City. I have no critiques or suggestions. I'm not sure what the minimum character count is, but i'm pasting in the abstract to fulfill it.

Background The coronavirus disease 2019 (COVID-19) is a systemic entity that

frequently implies neurological features at presentation and complications during the

disease course. We aimed to describe the characteristics and predictors for developing

in-hospital neurological manifestations in a large cohort of hospitalized patients with

COVID-19 in Mexico City. Methods We analyzed records from consecutive adult

patients hospitalized from March 15 to June 30, 2020, with moderate to severe COVID-

19 confirmed by reverse transcription real-time polymerase chain reaction (rtRT-PCR)

for the severe acute respiratory syndrome coronavirus 2 (SARS-CoV-2). Neurological

syndromes were actively searched by a standardized structured questionnaire and

physical examination, confirmed by neuroimaging, neurophysiology of laboratory

analyses, as applicable. Results We studied 1,072 cases (65% men, mean age

53.2±13 years), 71 patients had pre-existing neurological diseases (diabetic

neuropathy: 17, epilepsy: 15, history of ischemic stroke: eight, migraine: six, multiple

sclerosis: one, Parkinson disease: one), and 163 (15.2%) developed a new

neurological complication. Headache (41.7%), myalgia (38.5%), dysgeusia (8%), and

anosmia (7%) were the most common neurological symptoms at hospital presentation.

Delirium (13.1%), objective limb weakness (5.1%), and delayed recovery of mental

status after sedation withdrawal (2.5%), were the most common new neurological

syndromes. Age, headache at presentation, preexisting neurological disease, invasive

mechanical ventilation, and neutrophil/lymphocyte ratio ≥9 were independent

predictors of new in-hospital neurological complications. Conclusions Even after

excluding initial clinical features and pre-existing comorbidities, new neurological

complications in hospitalized patients with COVID-19 are frequent and can be

predicted from clinical information at hospital admission.

6. PLOS authors have the option to publish the peer review history of their article (what does this mean?). If published, this will include your full peer review and any attached files.

Reviewer #1: No

Reviewer #2: No

---

## [Author Response · Author response to Decision Letter 0]

19 Jan 2021

AUTHOR’S REVISION LETTER

Revision 1

We have analyzed the reviewers’ suggestions to our previous paper. We completely agree with these observations and have performed the changes in our new manuscript accordingly. Changes that apply in this revised version are highlighted. 

We profoundly appreciate the kind attention that the reviewers and the editorial team have given to our manuscript.

Response to reviewers

Editorial team comments

Response: We have consulted the Ethics Committee and as long as personal identification data are removed, there is no problem with data sharing. 

Response: We will submit the proper database in the corresponding field.

Reviewer 1

Overall, a very nice & timely cross-sectional study that provides important insight in to the neurologic manifestations of the COVID-19. The manuscript is well written and precise. It certainly deserves publication.

I see no need for major revisions.

I have 2 suggestions for minor revisions: (1) I would recommend a more explicit statement of the study design be included in the methods section. That the study is a cross-sectional analysis is mentioned in the introduction but should be explicitly stated in the methods. (2) Although the comparison groups are evident from the Tables, it is not clear from the text of the manuscript. The addition of text in the methodology ("The study aims to describe differences in demographics, clinical presentation, and disease severity in those patients who experienced neurologic events and those who did not. For these purposes, we define patients with neurologic events as those who suffered delirium, delayed recovery...") or something similar. This addition will lend more clarity to the text.

Response (1): We agree with this suggestion and have made the changes accordingly.

Response (2): We completely agree. This amendment was already made in the new version of the paper

Also, note that on p.7 of the submission, "Data Collection" section, there is a missing comma between pulmonary and neurological. Neurologic (rather than neurological - meaning pertaining to the study of or discipline of neurology) would be the preferred usage.

Response: Thank you very much. We have corrected these errors in our revised paper.

Reviewer 2

The authors have done a nice job in describing the neurological manifestations of COVID19 in Mexico City. I have no critiques or suggestions. I'm not sure what the minimum character count is, but i'm pasting in the abstract to fulfill it. 

Response: Thank you. We will check for the words/character count of the Abstract in detail and will make the amendments accordingly.

---

## [Decision Letter · Decision Letter 1]

8 Feb 2021

Neurological manifestations in hospitalized patients with COVID-19 in Mexico City

PONE-D-20-39539R1

Dear Dr. Chiquete,

We’re pleased to inform you that your manuscript has been judged scientifically suitable for publication and will be formally accepted for publication once it meets all outstanding technical requirements.

Kind regards,

Tai-Heng Chen, M.D.

Academic Editor

PLOS ONE

Reviewers' comments:

Reviewer's Responses to Questions

**Comments to the Author**

1. If the authors have adequately addressed your comments raised in a previous round of review and you feel that this manuscript is now acceptable for publication, you may indicate that here to bypass the “Comments to the Author” section, enter your conflict of interest statement in the “Confidential to Editor” section, and submit your "Accept" recommendation.

Reviewer #1: All comments have been addressed

Reviewer #2: All comments have been addressed

2. Is the manuscript technically sound, and do the data support the conclusions?

Reviewer #1: Yes

Reviewer #2: (No Response)

3. Has the statistical analysis been performed appropriately and rigorously? 

Reviewer #1: Yes

Reviewer #2: (No Response)

4. Have the authors made all data underlying the findings in their manuscript fully available?

Reviewer #1: Yes

Reviewer #2: (No Response)

5. Is the manuscript presented in an intelligible fashion and written in standard English?

Reviewer #1: Yes

Reviewer #2: (No Response)

6. Review Comments to the Author

Reviewer #1: I have no further recommendations for revision. I am have no further explanation to provide regarding the questions above.

Reviewer #2: (No Response)

7. PLOS authors have the option to publish the peer review history of their article (what does this mean?). If published, this will include your full peer review and any attached files.

Reviewer #1: No

Reviewer #2: No

---

## [Editor Report · Acceptance letter]

29 Mar 2021

PONE-D-20-39539R1 

Neurologic manifestations in hospitalized patients with COVID-19 in Mexico City 

Dear Dr. Chiquete:

I'm pleased to inform you that your manuscript has been deemed suitable for publication in PLOS ONE. Congratulations! Your manuscript is now with our production department. 

Kind regards, 

on behalf of

Dr. Tai-Heng Chen 

Academic Editor

PLOS ONE